# Effects of Muscle Fatigue and Recovery on the Neuromuscular Network after an Intermittent Handgrip Fatigue Task: Spectral Analysis of Electroencephalography and Electromyography Signals

**DOI:** 10.3390/s23052440

**Published:** 2023-02-22

**Authors:** Lin-I Hsu, Kai-Wen Lim, Ying-Hui Lai, Chen-Sheng Chen, Li-Wei Chou

**Affiliations:** 1Department of Physical Therapy and Assistive Technology, National Yang Ming Chiao Tung University, Taipei 112304, Taiwan; 2Biomechanics Research Laboratory, Department of Medical Research, MacKay Memorial Hospital, Taipei 25160, Taiwan; 3Department of Biomedical Engineering, National Yang Ming Chiao Tung University, Taipei 112304, Taiwan; 4Medical Device Innovation & Translation Center, National Yang Ming Chiao Tung University, Taipei 112304, Taiwan

**Keywords:** muscle fatigue, recovery, electroencephalography, electromyography, median frequency, power spectral density, corticomuscular coherence, corticocortical coherence

## Abstract

Mechanisms underlying exercise-induced muscle fatigue and recovery are dependent on peripheral changes at the muscle level and improper control of motoneurons by the central nervous system. In this study, we analyzed the effects of muscle fatigue and recovery on the neuromuscular network through the spectral analysis of electroencephalography (EEG) and electromyography (EMG) signals. A total of 20 healthy right-handed volunteers performed an intermittent handgrip fatigue task. In the prefatigue, postfatigue, and postrecovery states, the participants contracted a handgrip dynamometer with sustained 30% maximal voluntary contractions (MVCs); EEG and EMG data were recorded. A considerable decrease was noted in EMG median frequency in the postfatigue state compared with the findings in other states. Furthermore, the EEG power spectral density of the right primary cortex exhibited a prominent increase in the gamma band. Muscle fatigue led to increases in the beta and gamma bands of contralateral and ipsilateral corticomuscular coherence, respectively. Moreover, a decrease was noted in corticocortical coherence between the bilateral primary motor cortices after muscle fatigue. EMG median frequency may serve as an indicator of muscle fatigue and recovery. Coherence analysis revealed that fatigue reduced the functional synchronization among bilateral motor areas but increased that between the cortex and muscle.

## 1. Introduction

Muscle fatigue is an exercise-induced reduction in the maximal voluntary contraction (MVC) of muscles. In addition to muscle-level peripheral changes, the failure of the central nervous system to properly control motoneurons results in muscle fatigue [1]. At the muscle level, the median frequency of electromyography (EMG) decreases during muscle fatigue [2,3,4]. At the central level, a magnetoencephalography (MEG) study reported that the magnitude of post-movement beta rebound (PMBR) was reported to increase following fatigue [5]. In addition, an electroencephalography (EEG)-based spectral analysis reported that the power spectrum changed in relation to graded fatigue levels [6]. During muscle fatigue, the cortical activity of the contralateral sensorimotor cortex either decreases or plateaus, whereas that of the ipsilateral sensorimotor cortex increases [7].

As a function of frequency, coherence helps to quantify the association between signal pairs (e.g., brain and muscle signals) [8]. Cortical involvement in a task can be estimated using coherence to assess the activation of the motor cortex and investigate the correlation between the motor cortex and muscle activities. Coherence between cortical regions (corticocortical coherence; CCC) and between cortical and muscle activities (corticomuscular coherence; CMC) varies as a function of cortical involvement in a task.

The level of synchronization between EEG and EMG frequencies may serve as a functional connection between the cortex and muscle during fatigue [9]. The weakening of corticomuscular coupling may represent a major neural mechanism underlying muscle fatigue; this reduces the strength of corticomuscular signal coupling in the beta frequency band [10].

Cortical connectivity between two cortical regions can be effectively estimated using coherence in EEG [8]. EEG coherence measures the level of synchronization between two brain areas in terms of EEG signals recorded at different sites of the scalp [11]. A high level of CCC indicates a high level of synchronization between two brain areas and vice versa [12]. Poortvliet et al. [13] demonstrated that force control tasks were consistently associated with higher levels of CCC in the beta band than position control tasks within and between hemispheres.

The effects of muscle fatigue and recovery on the motor cortex, muscle, and their relationship remain unclear. In addition, most studies have mainly discussed coherence in pre/post fatigue conditions, while recovery from fatigue in the cortex and muscle has rarely been investigated. Therefore, in the present study, we investigated muscle function, cortical involvement, and the correlation between the changes in cortical and muscle activities during muscle fatigue and recovery. We used the flexor digitorum superficialis (FDS) muscle to measure handgrip strength because the FDS is a prime finger flexor for grip forming [14]. Furthermore, surface EMG can be easily performed for the FDS, which is a superficial muscle. We collected surface EMG data of the FDS and EEG data of the primary motor cortices of the bilateral hemispheres—left primary motor cortex (LM1) and right primary motor cortex (RM1)—to explore the changes in individuals’ muscle median frequency, CMC, and CCC after a handgrip fatigue task.

## 2. Participants and Methods

### 2.1. Ethical Approval

This study was approved by the Institutional Review Board of National Yang Ming Chiao Tung University (approval number: YM110099E). All procedures performed in this study were in accordance with the ethical principles of the Declaration of Helsinki. Informed consent was obtained from all participants.

### 2.2. Participants

This study included healthy right-handed individuals with no history of neuromuscular or musculoskeletal disorders in the dominant upper limb. The participants were requested to avoid any strenuous exercise for 72 h before the assessments.

### 2.3. Recordings

#### 2.3.1. EMG Data

Bipolar surface EMG signals were recorded using surface electrodes (Kendall^TM^ 200; Cardinal Health, Dublin, OH, USA) with a 25 mm interelectrode distance. After cleansing the skin with an alcohol swab, the bipolar electrode was secured over the FDS of the dominant arm. A reference electrode was placed on the skin overlying the lateral epicondyle of the dominant side. The collected analog EMG signals and force data were preamplified with a gain of 1000, filtered using a 3–1000 Hz band-pass filter (Grass P511; AstroNova, Inc., West Warwick, RI, USA), and digitalized using an analog-to-digital converter (CED Power1401; Science Products GmbH, Hofheim, Germany). Digital data were recorded at a sampling rate of 1000 Hz by using the Spike2 data acquisition system (Science Products GmbH, Hofheim, Germany) and were exported to MATLAB (MathWorks, Natick, MA, USA). The EMG data were filtered using band-pass filters of 8–450 Hz and a band-stop filter of 59–61 Hz with a fourth-order Butterworth filter.

#### 2.3.2. EEG Data

In accordance with the international 10/20 positioning method [15], we used a 16-channel EEG system with gel-based active electrodes (actiCAP; Brain Products GmbH, Hofheim, Germany). Data were sampled at 1000 Hz and referenced to the electrode FCz. A blunt needle was carefully pushed to fill the electrodes with conducting gel to connect the recording surface of each electrode with the scalp. The impedance of the EEG channels was maintained at <10 kΩ. The C3 and C4 electrodes represented the LM1 and RM1, respectively; this was to evaluate the essential functions of sensorimotor integration and motor control [16].

### 2.4. Experimental Procedure

The participants sat comfortably on a chair with their feet supported, shoulders adducted and neutrally rotated, elbow flexed at 90°, forearm in a neutral position, and wrist extended between 0° and 30°. They positioned their dominant arms on a supporting table to hold a handgrip dynamometer (Biometric Ltd., Newport, UK) (Figure 1).

The grip position of the dynamometer was adjusted to individuals’ hand sizes. This position was maintained for all test conditions. We explored the participants’ EEG–EMG-based activation patterns during the handgrip fatigue task (Figure 2).

Before the experiment, EEG data were collected for 30 s under a resting condition. Then, the participants performed three sets of maximal isometric contractions with their dominant hand. Each contraction was maintained for 5 s, with a 1 min rest between trials, to measure their maximal handgrip strength. The highest force value among the three trials was considered to be the MVC_b_ value, which was used to calculate the target value during the sustained submaximal contraction task and dynamic fatigue task. Then participants rested for 5 min.

Following the assessment of maximal handgrip strength, the participants were instructed to perform a fatigue task and a recovery task. During the fatigue task, they performed three sets of consecutive submaximal isometric contractions (60% MVC_b_) and one set of current MVC with uninterrupted repetitions prompted using a metronome (1 beat per second). The fatigue task was terminated when the current MVC was <50% of the MVC_b_ for the three sets. The participants were unaware of the termination criteria of the fatigue task and received visual feedback from a force-tracing template displayed on a monitor. The time to task failure was defined as the time course from the onset of a contraction to the termination of the test. The procedure of the fatigue task was explained to the participants before the test, emphasizing the need for maximum effort. In addition, they were verbally motivated throughout the test. During the recovery task, the participants performed a 5 s MVC test, with 1 min rest between two trials. The recovery task was terminated when the MVC was recovered to 80% of the MVC_b_ or the task time exceeded 10 min.

The sustained submaximal contraction tasks were performed at the following three time points: prefatigue state (task performed before fatigue), postfatigue state (task performed after fatigue), and postrecovery state (task performed after recovery). The sustained submaximal contraction task involved three sets of isometric voluntary grips at 30% of MVC_b_. Each contraction was maintained for 30 s and without any rest between two contractions.

The EMG and EEG of the FDS were simultaneously recorded during the sustained submaximal contraction tasks. Median frequency was analyzed at sustained submaximal contraction. CMC and CCC were calculated based on the EMG and EEG signals of the bilateral primary motor cortices (C3 and C4 channels).

### 2.5. Data Analysis

#### 2.5.1. Median Frequency

Median frequency values were calculated in the prefatigue, postfatigue, and postrecovery states of the sustained submaximal contraction task for comparison using MATLAB. The values were calculated in 1 s segments during the sustained submaximal contraction task. To analyze EMG median frequency, the EMG signal was filtered using a Hamming window and processed with a Fourier transform. Median frequency is the frequency that separates a power spectrum into two parts of equal energy, and it can be calculated using the following formula [17]:$$\int_{f1}^{Fmedian}PS\left(f\right)\cdot df=\int_{Fmedian}^{f2}PS\left(f\right)\cdot df$$
where *PS*(*f*) is the surface EMG power spectrum and can be calculated using a Fourier transform. The bandwidth of the surface EMG is partly determined by *f*1 and *f*2 (*f*1 = 8 Hz and *f*2 = 450 Hz of the bandwidth).

#### 2.5.2. Power Spectral Density

Using EEG analysis software (Brain Vision Analyzer; Brain Products, Gilching, Germany), EEG noise signal was removed using a 3–50 Hz band-pass filter, and the EEG signal artifacts that resulted from eye blinks or movements were removed using an independent component analysis algorithm. Power spectral density (*PSD*) was computed following Welch’s method and was used to calculate the areas of the alpha (8–12 Hz), beta (13–30 Hz), and gamma (31–50 Hz) bands. *Relative PSD* can be calculated using the following formula:$$Relative\;PSD=\frac{PSD_{n}}{PSD_{total}}$$

For normalization, we divided the energy spectral density values of the alpha, beta, and gamma bands by the *PSD* value obtained in the rest period to calculate *Relative PSD*.

#### 2.5.3. Coherence Analysis

Coherence analysis is dependent on spectrum analysis. CMC and CCC values were calculated for C4 and FDS, C3 and FDS, and C3 and C4. Coherence can be calculated using the following formula [18]:$$\left|Cxy\left(f\right)\right|=\frac{{\left|Pxy\left(f\right)\right|}^{2}}{Pxx\left(f\right)\times Pyy\left(f\right)}$$
where *Pxy(f)* is the cross-spectrum density between *x* and *y*, *Pxx(f)* is the auto spectral density of *x*, and *Pyy(f)* is the auto spectral density of *y*.

Finally, the area of coherence exceeding the critical threshold was calculated to determine the confidence level, which was set at 95% and calculated using a method described by Rosenberg et al. [19]. The critical threshold value can be calculated using the following formula:$$CT=1-{\left(1-\frac{\alpha }{100}\right)}^{\frac{1}{\left(n-1\right)}}$$
where *n* is the epoch number and α is the 5% confidence level [19].

### 2.6. Statistical Analysis

One-way repeated measures analysis of variance was used to determine whether the changes in median frequency before and after fatigue and recovery were different. If the result was significant, the Bonferroni correction test was used for post hoc analysis. Friedman’s test was used to compare the EEG, *PSD*, CCC, and CMC between the prefatigue, postfatigue, and postrecovery states. If the results were significant, the Wilcoxon signed-rank test was used for post hoc analysis. Statistical significance was set at *p* < 0.05.

## 3. Results

This section is divided into subheadings. It should provide a concise and precise description of the experimental results, their interpretation, as well as the experimental conclusions that can be drawn.

### 3.1. Median Frequency

A total of 20 individuals (thirteen males and seven females; mean age, 25.3 ± 2.5 years) participated in this study. In the comparison between sustained submaximal contraction tasks, the median frequency in the postfatigue state was significantly lower than that in the prefatigue (*p* < 0.001) and postrecovery (*p* < 0.001) states. No significant differences were observed between the prefatigue and postrecovery states in terms of median frequency (Figure 3).

### 3.2. PSD

No significant differences were noted among the three sustained submaximal contraction tasks in terms of the C3 *PSD* in the alpha, beta, or gamma bands (Figure 4a). Nevertheless, a prominent increase (17%) was noted in the C4 *PSD* in the gamma band in the postfatigue state compared with the findings in the prefatigue state (Figure 4b).

### 3.3. Coherence

#### 3.3.1. CMC

The CMC between the C3 and FDS increased significantly in the postfatigue state in the beta band compared with the findings in the other states (*p* < 0.05); no significant differences were observed between the prefatigue and postrecovery states. By contrast, in the gamma band, we found no significant differences among the three sustained submaximal contraction tasks (Figure 5a). CMC between the C4 and FDS increased significantly in the gamma band in the postfatigue and postrecovery states compared with the finding in the prefatigue state (*p* < 0.05). By contrast, in the beta band, we found no significant differences among the three sustained submaximal contraction tasks (Figure 5b).

#### 3.3.2. CCC

The beta and gamma coherence of the C3 and C4 increased significantly in the postrecovery state compared with the finding in the postfatigue state (*p* < 0.05). However, no significant differences were noted between the prefatigue and postrecovery states (Figure 6).

## 4. Discussion

In the present study, we investigated the changes in muscles after a handgrip fatigue task and recovery, as well as cortical involvement. After the fatigue task, the results corresponding to the functional coupling between the cortex and agonist muscle exhibited less CCC in the beta and gamma bands between the bilateral primary cortices; CMC beta and gamma markedly increased in the contralateral and ipsilateral primary cortices, respectively.

### 4.1. Median Frequency of EMG

We evaluated median frequency through an intermittent fatigue task closely resembling various ADLs. Therefore, this task was physiologically relevant for assessing fatigue. Median frequency substantially decreased in the postfatigue state but increased in the postrecovery state. A reduction in muscle fiber conduction velocity is a cause of signal power spectrum shift toward relatively low frequencies [20]. This instant response of EMG median frequency to muscle fatigue and fatigue recovery could be used as an indicator, as previously suggested [21,22]. More importantly, our findings also demonstrated that EMG median frequency rebounded back to the higher frequency range, suggesting that EMG median frequency can also serve as an indicator of muscle fatigue recovery.

### 4.2. EEG PSD

Through a sustained 30% MVC task, we compared *PSD* among the three states. The results were similar among the three states in terms of the contralateral primary motor cortex. Specifically, the gamma band in the ipsilateral primary cortex exhibited a prominent increase in the postfatigue state, and a continuous upward trend was noted in the postrecovery state. The correlation between functional magnetic resonance imaging (fMRI) and EMG signals was significant in the bilateral primary cortex and FDS in the sustained submaximal voluntary contraction task [7]. Cortical activities in the contralateral primary cortex, which were measured using fMRI, increased sharply during the early contraction period but plateaued during the last period. The fMRI signals in the ipsilateral primary motor cortex exhibited a steady increase [7]. Taghizadeh et al. [23] demonstrated that in the postfatigue state, the ipsilateral brain hemisphere exhibits higher levels of activity than the contralateral hemisphere in the beta and gamma bands. This observation is consistent with that of our study in that we also observed a prominent increase in the gamma band in the ipsilateral hemispheres in the postfatigue state. The activated brain center shifted toward the ipsilateral hemisphere in the fatigue state compared with the observations under nonfatigue conditions [24]. In the present study, the changes in the EEG power spectrum varied between the submaximal voluntary contraction and MVC tasks. Liu et al. [25] analyzed data obtained through a maximal voluntary task and reported considerable decreases in the alpha and beta bands in the contralateral hemisphere and in the beta band in the ipsilateral hemisphere. Ulloa et al. [26] revealed that the power spectrum of the sensorimotor cortex shifts to a high frequency during dynamic contractions. Thus, muscle fatigue due to submaximal intermittent fatigue contraction may increase the *PSD* in the gamma band.

### 4.3. CMC

Our findings revealed a fatigue-induced increase in the beta coherence between the contralateral primary motor cortex (C3) and FDS, which indicates an increased neural drive to target muscles for alleviating the effects of peripheral inhibitory signals generated due to fatigue. This finding corroborates that of an earlier study [27]. Ushiyama et al. [28] found that the steadiness of the exerted force decreases with increasing beta coherence. This observation supports our finding obtained for participants in the postfatigue state. In the fatigue state, the participants were unstable when maintaining strength and could not stably maintain the target strength. Therefore, the intensity of the beta band was increased to maintain and control the strength output to the target strength. Tecchio et al. [9] analyzed the MEG-EMG coherence after muscle fatigue. Their finding echoes our results which showed that the coherence in the beta band was higher postfatigue than prefatigue. Conversely, Yang et al. [10] demonstrated that fatigue weakens the strength of brain–muscle signal coupling in the beta frequency band. Siemionow et al. [29] analyzed maximal intermittent handgrip contractions and found that beta and gamma coherence decreased during muscle fatigue. The differences between the present and previous studies [10,29] may be because of the differences in study design. FDS was selected as an agonist muscle during the handgrip task to avoid muscular compensation during muscle fatigue and thus believed to be more precisely measurable in terms of CMC. Furthermore, central fatigue contributes proportionally more to the total force reduction during sustained submaximal contractions than during maximal contractions [30]. Thus, our study, in which we included a submaximal intermittent fatigue task, led to different findings compared with those reported by Siemionow et al. [29].

The increase in the gamma band of the CMC between the ipsilateral primary motor cortex (C4) and FDS in fatigue and recovery indicates that the ipsilateral primary motor cortex helps enhance the functional connection between the central nervous system and muscle. This phenomenon has been explained by Liu et al. [24], who demonstrated that the center of brain activation exhibited substantial location shifts during a fatigue motor task. During fatigue, the brain may avoid fatigue by shifting neuron populations that participate in a fatigue motor task [31], thus increasing the functional connection between the cortex and muscle.

### 4.4. CCC

High CCC indicates an increased functional synchronization among two or more brain areas associated with preparations for task execution [32]. Classen et al. [33] reported that the beta band of the CCC between the motor and visual regions of the cortex is greater during visuomotor tasks than during a motor task without any visual feedback. We found that beta (13–30 Hz) and gamma (31–50 Hz) band coherence of the bilateral primary motor cortex (C3 and C4) decreased after the intermittent handgrip fatigue task but gradually recovered in the postrecovery state. This indicates relatively low levels of cortical network involvement in the postfatigue state; high levels of cortical involvement were observed between the LM1 and RM1. Similar findings were reported by Selenia et al. [34], who demonstrated higher levels of coherence at rest than those after time-to-exhaustion trials in an endurance cycling task. After the exertion of the maximum effort, a counteracting mechanism restores coherence during recovery with the same pattern of functional connectivity among brain areas as that noted during rest. Notably, Selenia et al. [34] also found that the focus of attention on muscle exertion that leads to an increased coherence of the bilateral primary motor cortex (C3–C4) was devoted to the motor input in the bilateral cycling task. CCC findings obtained in the present study may facilitate clinical application to optimize motor performance.

### 4.5. CMC and CCC Applications

The study of the cortical–muscle and cortical–cortical functional coupling can help better understand how the cerebral cortex controls muscle and how the bilateral sensorimotor cortices interact with each other [18]. Thus, the current findings of CMC and CCC can serve as good indicators of central fatigue and fatigue recovery, which is beneficial for individuals with fatigue syndromes in the rehabilitation clinic [35,36], or even athletes in training or competition.

## 5. Conclusions

EMG median frequency may serve as an indicator of muscle fatigue and recovery. The coherence analysis revealed that muscle fatigue reduces coherence in the bilateral primary cortex but increases that between the primary motor cortex and FDS. Fatigue reduces functional synchronization among motor areas but enhances it between the cortex and muscle. In the postfatigue state, increases are noted in the beta and gamma bands of the CMC between the contralateral sensorimotor cortex and muscle and between the ipsilateral sensorimotor cortex and muscle, respectively; this helps improve the functional connection between the central nervous system and muscle.

## Figures and Tables

**Figure 1 sensors-23-02440-f001:**
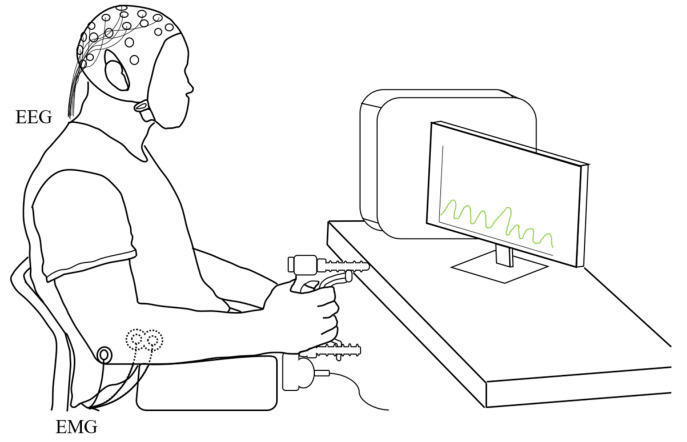
Experimental setup and posture of the participants.

**Figure 2 sensors-23-02440-f002:**
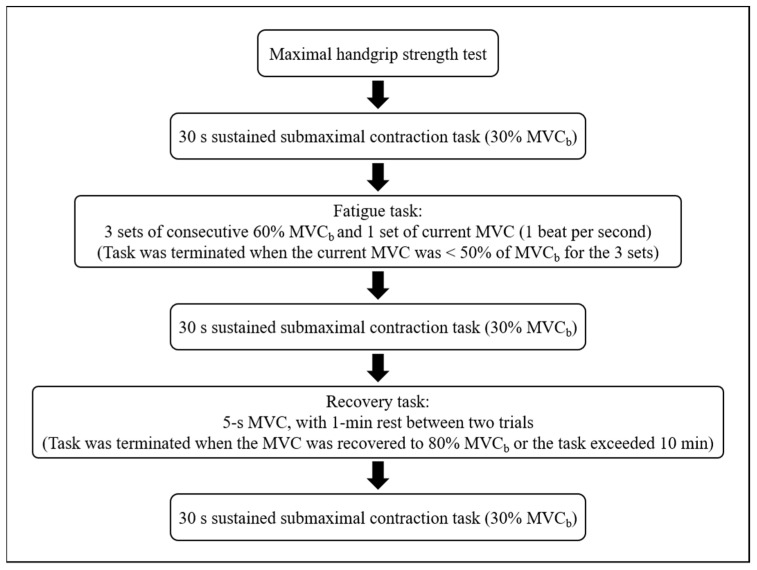
Experimental process.

**Figure 3 sensors-23-02440-f003:**
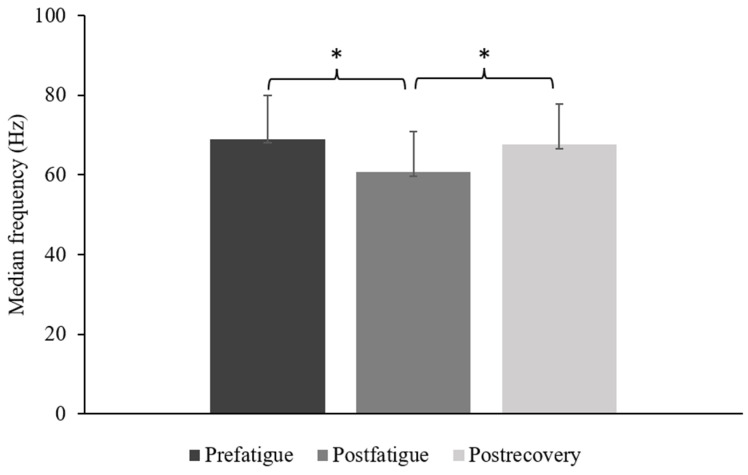
Comparison of the FDS median frequency among the prefatigue, postfatigue, and postrecovery states. (* *p* < 0.05).

**Figure 4 sensors-23-02440-f004:**
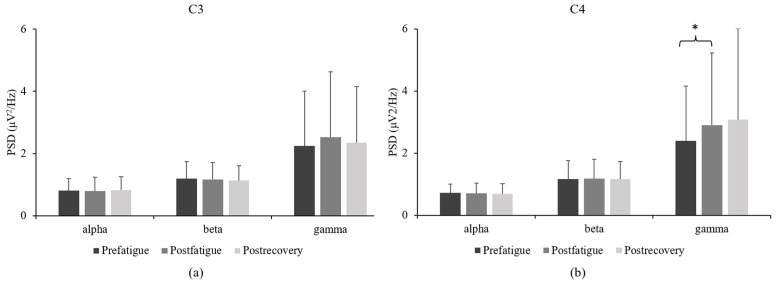
Comparison of (**a**) C3 power spectrum density and (**b**) C4 power spectrum density among the prefatigue, postfatigue, and postrecovery states. (* *p* < 0.05).

**Figure 5 sensors-23-02440-f005:**
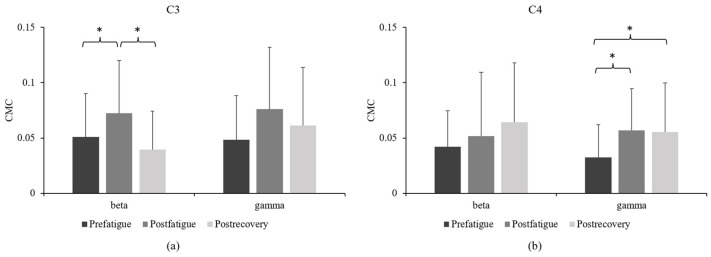
Comparison of (**a**) C3 corticomuscular coherence and (**b**) C4 corticomuscular coherence among the prefatigue, postfatigue, and postrecovery states. (* *p* < 0.05).

**Figure 6 sensors-23-02440-f006:**
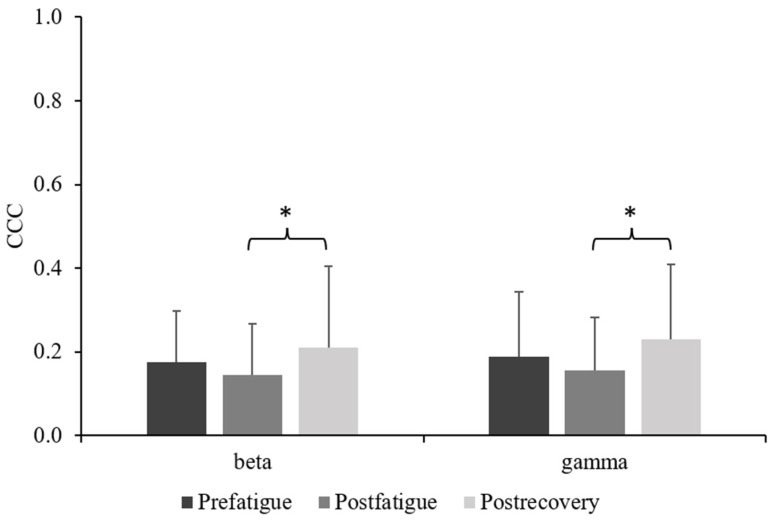
Comparison of C3–C4 corticocortical coherence among the prefatigue, postfatigue, and postrecovery states. (* *p* < 0.05).

## Data Availability

The data presented in this study are available on request from the corresponding author. The data are not publicly available due to ethical issue.

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
