# Peer review of "Effects of Muscle Fatigue and Recovery on the Neuromuscular Network after an Intermittent Handgrip Fatigue Task: Spectral Analysis of Electroencephalography and Electromyography Signals"

_sensors, 2023, doi:10.3390/s23052440_

Round 1

Reviewer 1 Report

RE: 2164203

Title: Effects of Muscle Fatigue and Recovery on the Neuromuscular Network After an Intermittent Handgrip Fatigue Task:  Spectral Analyses of Electroencephalography and Electromyography Signals.

To the Author(s)

The effect of muscle fatigue and recovery has been of interest over the past decade and still is. Spectral analyses Electroencephalography (EEG) and Electromyography (EMG) Signals is the main contribution of this paper.  The procedure is well described. Results are clearly discussed and good representative figures are given which serves to the conclusion that EMG median frequency may serve as an indicator of muscle fatigue and recovery. Moreover, the authors deduce that improving functional connection between central nervous system and muscle may be through the beta  band of the CMC between the contralateral sensorimotor cortex and muscle as well as the gamma band of the CMC between the ipsilateral sensorimotor cortex and muscle.

Linguistically. The paper is clear and well written. Please preferably change the word Analyses to Analysis starting from the title to all through.  Although there are different tests they are all spectral and hence we believe the singular form would be more appropriate.  This is certainly dependent on whether  the journal allows a modification to the title.

Technically, we recommend to authors to include a review of the following article by  researchers in the field. The reason is that the article did not use the most recent machine learning techniques which more advanced techniques may be used and the article also mentions the need for a better indicator.  Thence, the paper at hand shows that the median of frequency may serve as the required indicator.

“Effect of Muscle Fatigue on surface Electromyography – Based Hand Grasp Force estimation”

By

Jinfeug Wang, Muye Pang, Peixuan Yu, Biwei Tang, Kui Xiang and Zhaojie Ju.

Published in Appplied Bionics and Biomechanics

Special Issue Volune 2021 . Open Access. Article ID 8817480.

https://doi.org/10.1155/2021/8817480

Reviewer 2 Report

The authors proposed a method to investigate the effects of muscle fatigue and recovery on the neuromuscular network using very straightforward metrics such as median frequency for FDS, PSD for EEG, CCC, and CMC on 20 healthy right-handed volunteers.

The paper is written well, and they did propose adequate justification for their chosen 3 channels for EEG, one channel EMG.

However, there are some major concerns about this paper:

1- I do not see that they try to first address any gap existing in the literature of cortico-muscular coherency study and propose a solution for that. I suggest they explain why their approach is novel and different from other papers in this area.

2- They have not tried to connect the findings of their paper to any potential application in the areas of corrective exercise or rehabilitation research or so on. So, it is not clear to me how the findings of their paper would help any reader.

 3- I encourage the authors to explain how their results are possibly not related to the individual athletic experience of each participant. It is very common that we see a large amount of inter-subject variability and this would need a discussion. 

Reviewer 3 Report

This manuscript reports the analyzed the effects of muscle fatigue and recovery on the neuromuscular network through the spectral analyses of EEG and EMG. The coherence of EEG and EMG are analyzed. The paper is completed in the present form. The reviewer recommends publishing it in the journal.

Round 2

Reviewer 2 Report

I think they have done a good job of responding to my comments.